# Radiation of High-Energy Gamma Quanta by Ultrarelativistic Electrons on Nuclei in Strong X-ray Fields

**Sergei Roshchupkin** [1,*], **Alexander Dubov** [1,2,†] **and Stanislav Starodub** [3,†]

1.  Higher School of Fundamental Physical Research, Peter the Great St. Petersburg Polytechnic University, 195251 St. Petersburg, Russia; alexanderpolytech@gmail.com
2.  Department of Applied Physics, Aalto University, 02150 Espoo, Finland
3.  Institute of Applied Physics, National Academy of Sciences of Ukraine, 40000 Sumy, Ukraine; starodubss@gmail.com
*   Correspondence: serg9rsp@gmail.com
†   These authors contributed equally to this work.

**Abstract:** The possibility of radiation of high-energy gamma quanta with energies of the order of $100\,\mathrm{GeV}$ by ultrarelativistic electrons on nuclei in strong X-ray fields with intensities up to $\sim\!10^{27}\,\mathrm{Wcm^{-2}}$ was theoretically studied. It is shown that this effect can be realized under special experimental conditions in the process of resonant spontaneous bremsstrahlung radiation of ultrarelativistic electrons on nuclei in an external electromagnetic field. These special experimental conditions determine the characteristic energy of the electrons. This characteristic energy should be significantly less than the energy of the initial electrons. Under these conditions, spontaneous gamma quanta are emitted in a narrow cone with energies close to the energy of the initial electrons. Moreover, the resonant differential cross-sections of such processes can exceed the corresponding differential cross-section without an external field by twenty orders of magnitude. The results obtained can explain the occurrence of high-energy gamma quanta near pulsars and magnetars.

**Keywords:** open quantum systems; quantum electrodynamics of strong X-ray fields; ultrarelativistic electrons; high-energy spontaneous gamma quanta; nuclei; strong X-ray fields of pulsars and magnetars; Oleinik resonances

## 1. Introduction

Astroparticle physics has been attracting attention for a long time (see, for example, articles [1–4]). At the same time, the processes of quantum electrodynamics (QED) in X-ray fields near pulsars and magnetars are intensively studied [5,6]. We also note a fairly large number of articles on QED processes in strong electromagnetic fields (see, for example, articles [7–42]). It is important to emphasize that higher-order QED processes with respect to the fine structure constant in an electromagnetic field can proceed both in a resonant and non-resonant way. Here, the so-called Oleinik resonances may occur [7,8], due to the fact that lower-order processes are allowed in the laser field by the fine structure constant. It is important to note that the resonant differential cross-section can significantly exceed the corresponding non-resonant one [19–22,26,34–36].

Resonant spontaneous bremsstrahlung (SB) radiation of ultrarelativistic electrons by nuclei in an electromagnetic field was studied in articles [22,34,35]. In the article of [22], this process was considered in a strong electromagnetic field of optical frequencies; however, these articles did not study the case of strong fields of the X-ray frequency range, when spontaneous gamma quanta are emitted with energies close to the energies of the initial electrons. It is this, the most interesting case, that is considered in this article.

We studied the resonance SB process for high-energy particles when the basic classical parameter $\eta$ satisfies the relation

$$\eta \ll \frac{E_{i,f}}{mc^2} \gg 1 \qquad \eta = \frac{eF\lambdabar}{mc^2}. \tag{1}$$

Here $e$ and $m$ are the charge and mass of the electron, $F$ and $\lambdabar = c/\omega$ are the electric field strength and wavelength, $\omega$ is the frequency of the wave, and $E_i$ and $E_f$ are the energy of the initial and final electrons.

In the article of [22], it was shown that the resonant frequency of a spontaneous gamma quantum is determined by its outgoing angle relative to the initial electron momentum of the (for channel A, see Figure 1A) or the final electron momentum (for channel B, see Figure 1B), as well as the quantum parameter

$$\kappa_{\eta(r)} = \frac{r}{r_\eta}. \tag{2}$$

Here the parameter $\kappa_{\eta(r)}$ is numerically equal to the ratio of the number of absorbed photons of the wave in the laser-stimulated Compton process ($r \geq 1, 2, 3 \dots$) to the characteristic quantum parameter $r_\eta$, which is determined by the experimental conditions and the laser installation.

$$r_\eta = \frac{\left(mc^2\right)^2 \left(1 + \eta^2\right)}{4(\hbar\omega)E_i \sin^2(\theta_i/2)}. \tag{3}$$

where $\theta_i$ is the angle between the momentum of the initial electron and the direction of wave propagation. It can be seen from expression (3) that the value of the characteristic parameter $r_\eta$ is inversely proportional to the photon energy of the wave ($\hbar\omega$) and the energy of the initial electrons ($E_i$), and is also directly proportional to the intensity of the wave $\left(I \sim \eta^2 \left(\text{Wcm}^{-2}\right)\right)$. It is important to note that the value of the parameter $\kappa_{\eta(r)}$ significantly affects the resonant frequency of the spontaneous gamma quantum. So, for $\kappa_{\eta(r)} \ll 1$ the resonant frequency is much less than the energy of the initial electron $\left(\hbar\omega'_{\eta(r)} \sim \kappa_{\eta(r)} E_i \ll E_i\right)$. If a parameter $\kappa_{\eta(r)} \sim 1$, then the resonant frequency is of the same order with the energy of the initial electron $\left(\hbar\omega'_{\eta(r)} \sim E_i\right)$. If condition $\kappa_{\eta(r)} \gg 1$ is met, then the resonant frequency will be close to the energy of the initial electron $\left(\hbar\omega'_{\eta(r)} \to E_i\right)$. At the same time, for a fixed value of the parameter $r_\eta$, the processes with a small number of absorbed photons of the wave ($r \sim 1$) have the largest resonant cross-section. As the number of absorbed photons of the wave increases ($r \gg 1$), the resonant cross-section decreases rapidly. It is important to emphasize that the production of high-energy spontaneous gamma quanta is possible only under the condition

$$\kappa_{\eta(r)} \gg 1. \tag{4}$$

The fulfillment of condition (4) is possible in two areas of variation of the quantum parameter $r_\eta$ (3). Thus, for $r_\eta \gtrsim 1$, condition (4) is fulfilled only for a large number of absorbed photons of the wave $r \gg 1$, when the probability of these processes is significantly less than the probability of the emission of low-energy spontaneous gamma quanta. This case was studied in detail in [22]. On the other hand, when

$$r_\eta \ll 1 \tag{5}$$

the fulfillment of condition (4) already takes place for a small number of absorbed photons of the wave ($r \geq 1, 2, 3 \dots$), when the probability of these processes will be the maximum. This case was not considered.

In this paper, we study the case of radiation of narrow high-energy gamma-ray beams under appropriate experimental conditions for a quantum parameter $r_\eta$ (5).

We use the relativistic system of units: $\hbar = c = 1$.

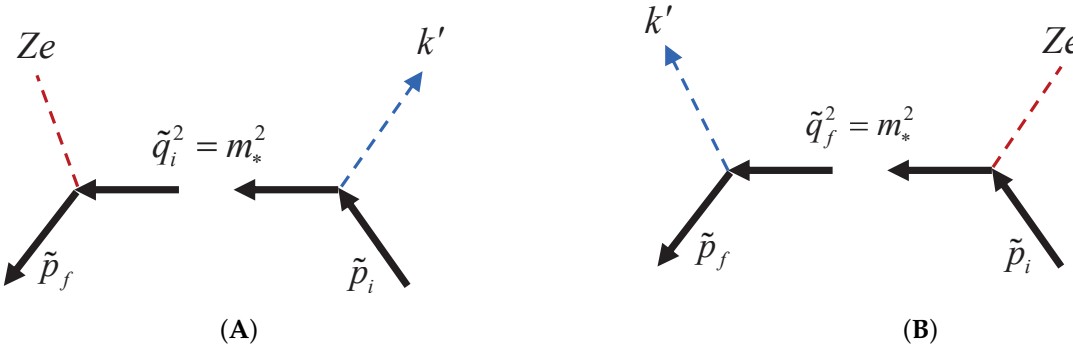

**Figure 1.** Resonant spontaneous bremsstrahlung of an electron in the field of a nucleus and a plane electromagnetic wave. Reaction channels (**A**,**B**).

## 2. Resonant Frequency in a Strong Field

Oleinik resonances occur when an intermediate electron in the electromagnetic wave field enters the mass shell [22,34,35]. Because of this, for channels A and B, we obtain:

$$\tilde{q}_i^2 = m_*^2, \; \tilde{q}_i = \tilde{p}_i + rk - k'. \tag{6}$$

$$\tilde{q}_f^2 = m_*^2, \; \tilde{q}_f = \tilde{p}_f + k' - rk. \tag{7}$$

Here $\tilde{q}_i$ and $\tilde{q}_f$ are the 4-quasimomenta of intermediate electrons for channels A and B, $m_*$ is the effective mass of the electron in the plane wave field

$$\tilde{p}_j = p_j + \eta^2 \frac{m^2}{2(kp_j)} k, \; \tilde{q}_j = q_j + \eta^2 \frac{m^2}{2(kq_j)} k, \quad j = i, f. \tag{8}$$

$$\tilde{p}_{i,f}^2 = m_*^2, \; m_* = m\sqrt{1 + \eta^2}, \quad kp_j = \omega E_j - \mathbf{k}\mathbf{p}_j. \tag{9}$$

In expressions (6)–(8) $k = (\omega, \mathbf{k})$ and $k' = (\omega', \mathbf{k}')$ are the 4-momenta of the external field photon and spontaneous gamma quantum.

We studied the most interesting case of ultrarelativistic electron energies, when the spontaneous gamma quantum and the final electron fly out in a narrow cone along the momentum of the initial electron. In this case, the direction of the propagation of the wave lies far from the given narrow cone of particles (if the direction of propagation of the wave lies inside the narrow cone of particles, then the resonances disappear.

$$E_{i,f} >> m, \tag{10}$$

$$\theta'_{i,f} = \angle\left(\mathbf{k}', \mathbf{p}_{i,f}\right) << 1, \quad \theta_{if} = \angle\left(\mathbf{p}_i, \mathbf{p}_f\right) << 1, \tag{11}$$

$$\theta' = \angle(\mathbf{k}', \mathbf{k}) \sim 1, \quad \theta_{i,f} = \angle\left(\mathbf{k}, \mathbf{p}_{i,f}\right) \sim 1. \tag{12}$$

In this paper, we consider the energies of the initial electrons $E_i \lesssim 10^3$ GeV and also in a wide range of photon energies of an electromagnetic wave $\left(1 \text{ eV} \lesssim \omega \lesssim 10^4 \text{ eV}\right)$. At the same time, we consider the intensities of the electromagnetic wave to be significantly less than the critical intensities of the Schwinger $\left(I \ll I_* \sim 10^{29} \text{ Wcm}^{-2}\right)$.

We determine the resonant frequency $\left(\omega'_{\eta i(r)}\right)$ of the spontaneous gamma quantum for channel A (see Figure 1A).

We take into account the relations (10)–(12) in the resonant condition (6). After simple calculations, we obtain

$$x'_{\eta i(r)} = \left[1 + \frac{\left(1 + \delta'^2_{\eta i}\right)}{\kappa_{\eta(r)}}\right]^{-1}, \quad x'_{\eta i(r)} = \frac{\omega'_{\eta i(r)}}{E_i}. \tag{13}$$

Here it is indicated:

$$\delta'_{\eta i} = \frac{E_i \theta'_i}{m_*} = \frac{E_i \theta'_i}{m \sqrt{1 + \eta^2}}. \tag{14}$$

It can be seen from expression (13) that the resonant frequency of a spontaneous gamma quantum is determined by the outgoing angle (ultrarelativistic parameter $\delta'_{\eta i}$), as well as the quantum parameter $\kappa_{\eta(r)}$ (2), (3). Note that the resonant radiation spectrum (13) is essentially discrete, since each value of the number of absorbed laser photons corresponds to its own resonant frequency of the spontaneous gamma quantum: $r \to \omega'_{\eta i(r)}$ (13). The resonance spectrum of spontaneous radiation in the region $r_\eta \gtrsim 1 \left(0 < \kappa_{\eta(r)} < \infty\right)$ was studied in detail in the article [22].

Here we study the case when the quantum characteristic parameter $r_\eta$ satisfies condition (5). Under these conditions, the parameter $\kappa_{\eta(r)}$ will satisfy condition (4). As a result, for not very large outgoing angles $\left(\delta'^2_{\eta i} \ll \kappa_{\eta(r)}\right)$ for channel A the resonant frequency of the spontaneous gamma quantum (13) will be close to the energy of the initial electron

$$x'_{\eta i(r)} \approx 1 - \frac{\left(1 + \delta'^2_{\eta i}\right)}{\kappa_{\eta(r)}} \approx 1 \left(\delta'^2_{\eta i} \ll \kappa_{\eta(r)} \gg 1\right). \tag{15}$$

Thus, in this case, with a small number of absorbed photons of the wave, the resonant spontaneous gamma quanta of the maximum frequency are emitted. The condition (5) can be rewritten for the energies of the initial electrons

$$E_i \gg E_*, \qquad E_* = \frac{m^2 \left(1 + \eta^2\right)}{4\omega \sin^2 \left(\theta_i / 2\right)}. \tag{16}$$

Then, for the characteristic energy $E_*$ when the electron beam moves towards the electromagnetic wave $(\theta_i = \pi)$ and different frequencies and intensities of the wave, we obtain:

$$E_* = 6.5 \times 10^{10} \frac{\left(1 + \eta^2\right)}{\omega(\text{eV})} \text{ eV} = \begin{cases} 0.65 \text{ GeV}, & \text{if } \omega = 0.2 \text{ keV}, I = 0.746 \times 10^{23} \text{ Wcm}^{-2} (\eta = 1) \\ 6.5 \text{ MeV}, & \text{if } \omega = 20 \text{ keV}, I = 0.746 \times 10^{27} \text{ Wcm}^{-2} (\eta = 1) \end{cases}. \tag{17}$$

Consequently, for electron energies satisfying conditions (16), (17) resonant spontaneous gamma quanta will be emitted with energies close to the energies of the initial electrons (15).

We obtain the equation for the resonant frequency $\omega'_{\eta f(r)}$ of the spontaneous gamma quantum in the case of channel B (see Figure 1B). Given the kinematics Equations (10)–(12), from the expression (7) we obtain:

$$\delta'^2_{\eta f} x'^3_{\eta f(r)} - 2\delta'^2_{\eta f} x'^2_{\eta f(r)} + \left(1 + \delta'^2_{\eta f} + \kappa_{\eta(r)}\right) x'_{\eta f(r)} - \kappa_{\eta(r)} = 0, \quad x'_{\eta f(r)} = \frac{\omega'_{\eta f(r)}}{E_i}. \tag{18}$$

Here it is indicated

$$\delta'_{\eta f} = \frac{E_i \theta'_f}{m_*} = \frac{E_i \theta'_f}{m \sqrt{1 + \eta^2}}. \tag{19}$$

A detailed study of Equation (18) for the case $r_\eta \gtrsim 1$ $\left(0 < \kappa_{\eta(r)} < \infty\right)$ was carried out in [22]. Here, we study the case when the quantum characteristic parameter $r_\eta$ satisfies condition (5). Under these conditions, the parameter $\kappa_{\eta(r)}$ will satisfy condition (4). As a result, for not very large outgoing angles $\left(\delta'^2_{\eta i} \ll \kappa_{\eta(r)}\right)$, from Equation (18), it is easy to obtain the resonant frequency of a spontaneous gamma quantum for channel B

$$x'_{\eta f(r)} \approx 1 - \frac{\left(1 + \delta'^2_{\eta f}\right)}{\kappa_{\eta(r)}} \approx 1 \left(\delta'^2_{\eta f} \ll \kappa_{\eta(r)} \gg 1\right). \tag{20}$$

## 3. The Maximum Resonant Differential SB Cross-Sections

A general relativistic expression for a resonant differential cross-section with simultaneous registration of the outgoing angle and frequency of the spontaneous gamma quantum for channels A and B was obtained in the article [22]. This expression has been studied in detail in the range of quantum parameter values $r_\eta \gtrsim 1$; however, the most interesting area of these parameter values has not been studied $r_\eta \ll 1$ (5).

Note that the probability of a resonant differential cross-section of spontaneous bremsstrahlung is obtained from the corresponding amplitude of the process in a standard way. At the same time, the corresponding probability of the process is multiplied by the concentration of nuclei $n_i$. When a resonant differential cross-section is obtained, the corresponding probability of the process is divided by the flux $v_i n_i = |\mathbf{p}_i/E_i| \cdot n_i$. Because of this, the resonant cross-section does not depend on the concentration of nuclei $n_i$ [43].

Taking into account the parameters of the electromagnetic wave (5) and the energies of the initial electrons (16), as well as expressions for the resonant frequencies (15) and (20), we obtain expressions for the maximum resonant differential cross-section for channels A and B:

$$R^{\max}_{\eta i(r)} = \frac{d\sigma^{\max}_{\eta i(r)}}{dx'_{\eta i(r)} d\delta'^2_{\eta i}} = \left(Z^2 \alpha r_e^2\right) c_{\eta i} \Psi_{\eta i(r)}, \tag{21}$$

$$R^{\max}_{\eta f(r)} = \frac{d\sigma^{\max}_{\eta f(r)}}{dx'_{\eta f(r)} d\delta'^2_{\eta f}} = \left(Z^2 \alpha r_e^2\right) c_{\eta i} \Psi_{\eta f(r)}, \tag{22}$$

Here, $\alpha$ is the fine structure constant, $Z$ is the charge of the nucleus, $r_e$ is the classical radius of the electron, the $\Psi_{\eta i(r)}$ and $\Psi_{\eta f(r)}$ functions determine the spectral-angular distribution of the resonant SB cross-section for channels A and B:

$$\Psi_{\eta i(r)} = \frac{\kappa_{\eta(r)}}{\left(1 + \delta'^2_{\eta i}\right) g_{\eta i(r)}} K\left(u_{\eta i(r)}, \kappa_{\eta(r)}\right), \tag{23}$$

$$\Psi_{\eta f(r)} = \frac{\left(1 + \delta'^2_{\eta f}\right)}{\kappa_{\eta(r)}} g_{\eta f(r)} K\left(u_{\eta f(r)}, \kappa_{\eta(r)}\right), \tag{24}$$

and $c_{\eta i}$ is the coefficient, which is determined by the parameters of the laser installation

$$c_{\eta i} = \pi \left(\frac{8\pi^2 E_i}{\alpha m_* K(r_\eta)}\right)^2 \approx 3.67 \times 10^8 \frac{E_i^2}{m^2 (1 + \eta^2) K^2(r_\eta)} \gg 1. \tag{25}$$

In expressions (23) and (24), the relativistic-invariant parameters $u_{\eta i(r)}$ and $u_{\eta f(r)}$ have the form:

$$u_{\eta j(r)} \approx \frac{\kappa_{\eta(r)}}{\left(1 + \delta'^2_{\eta j}\right)}, \qquad j = i, f. \tag{26}$$

The function $K(r_\eta)$ in expression (25) is determined by the resonance width (the full probability of the Compton effect stimulated by the electromagnetic field) and has the form [31]:

$$K(r_\eta) = \sum_{r=1}^{\infty} K_r(r_\eta), \quad K_r(r_\eta) = \int_{0}^{r/r_\eta} \frac{du}{(1+u)^2} K(u, \kappa_{\eta(r)}). \tag{27}$$

$$K(u, \kappa_{\eta(r)}) = -4J_r^2(\gamma_{\eta(r)}) + \eta^2\left(2 + \frac{u^2}{1+u}\right)(J_{r+1}^2 + J_{r-1}^2 - 2J_r^2), \tag{28}$$

$$\gamma_{\eta(r)} = 2r\frac{\eta}{\sqrt{1+\eta^2}}\sqrt{\frac{u}{\kappa_{\eta(r)}}\left(1 - \frac{u}{\kappa_{\eta(r)}}\right)}, \tag{29}$$

In expressions (23) and (24), functions $K(u_{\eta i(r)}, \kappa_{\eta(r)})$ and $K(u_{\eta f(r)}, \kappa_{\eta(r)})$ are obtained from ratios (28), (29) by corresponding substitutions: $u \to u_{\eta i(r)}$ and $u \to u_{\eta f(r)}$ (26). The $g_{\eta i(r)}$, $g_{\eta f(r)}$ functions determine the square of the momentum transmitted to the nucleus, taking into account the relativistic corrections of the order $m_*^2/E_i^2$ for channels A and B. So, for channel A, we obtain:

$$g_{\eta i(r)} = g_{\eta i(r)}^{(0)} + \frac{1}{(1+\eta^2)}g_{\eta i(r)}^{(1)} + \frac{1}{(1+\eta^2)^2}g_{\eta i(r)}^{(2)}, \tag{30}$$

$$g_{\eta i(r)}^{(0)} \approx \frac{\delta_{\eta i}^{\prime 4}\kappa_{\eta(r)}^3}{6\left(1+\delta_{\eta i}^{\prime 2}\right)^3} + \frac{\beta_{\eta(r)}^{\prime}}{2\sin^2(\theta_i/2)}\left[\beta_{\eta(r)}^{\prime} - \frac{2\kappa_{\eta(r)}^2}{\left(1+\delta_{\eta i}^{\prime 2}\right)^2}\delta_{\eta i}^{\prime 2}\right], \tag{31}$$

$$g_{\eta i(r)}^{(1)} \approx \frac{\kappa_{\eta(r)}}{\left(1+\delta_{\eta i}^{\prime 2}\right)}\left[\beta_{\eta(r)}^{\prime} + \frac{\kappa_{\eta(r)}}{\left(1+\delta_{\eta i}^{\prime 2}\right)}\delta_{\eta i}^{\prime 2}\right], \quad g_{\eta i(r)}^{(2)} \approx \frac{\kappa_{\eta(r)}^2}{2\left(1+\delta_{\eta i}^{\prime 2}\right)^2}, \tag{32}$$

$$\beta_{\eta(r)}^{\prime} \approx \kappa_{\eta(r)}\left[\left(\frac{\eta^2}{1+\eta^2}\right)\frac{1}{\left(1+\delta_{\eta i}^{\prime 2}\right)} - 1\right]. \tag{33}$$

Note that for channel B, the $g_{\eta f(r)}$ functions are obtained from $g_{\eta i(r)}$ functions (30)–(33) by replacing: $\delta_{\eta i}^{\prime} \to \delta_{\eta f}^{\prime}\left(1 + \delta_{\eta f}^{\prime 2}\right)/\kappa_{\eta(r)}$.

It is important to emphasize that the obtained expressions (21), (22) are appropriate for the case of a single initial electron. To derive the relations for the case of an electron flux, one has to multiply the corresponding equations by the concentration $n_e$.

A comparison of the resonant differential cross-sections for channels A (21), (23) and B (22), (24) shows that channel A is dominant

$$\frac{R_{\eta i(r)}^{\max}}{R_{\eta f(r)}^{\max}} = \frac{\Psi_{\eta i(r)}}{\Psi_{\eta f(r)}} \gtrsim \kappa_{\eta(r)} \gg 1. \tag{34}$$

## 4. Results

Figures 2 and 3 show the distributions of the maximum resonant differential cross-section for channels A (21) and B (22) from the value of the square of the outgoing angle of the spontaneous gamma quantum with a different number of absorbed photons of the wave.

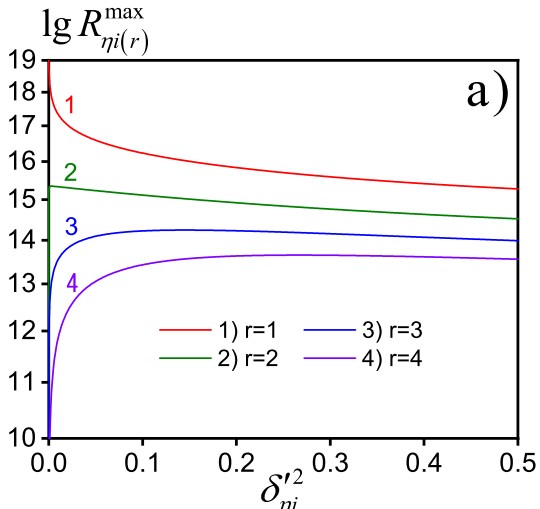
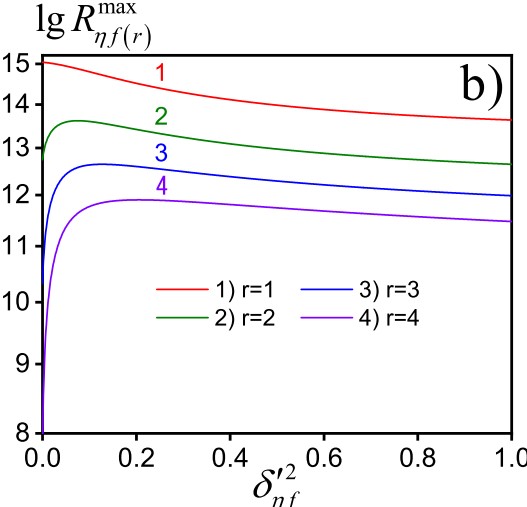

**Figure 2.** The maximum resonant differential cross-section (in units of $Z^2\alpha r_e^2$) as functions of the corresponding outgoing angles, plotted for the different values of absorbed wave photons. (**a**) $R^{\max}_{\eta i(r)}$ (21) for channel A. (**b**) $R^{\max}_{\eta f(r)}$ (22) for channel B. The frequency and intensity of the electromagnetic wave are $\omega = 0.2$ keV and $I = 0.746 \times 10^{23}$ Wcm$^{-2}$. The energy of the initial electrons is $E_i = 65$ GeV.

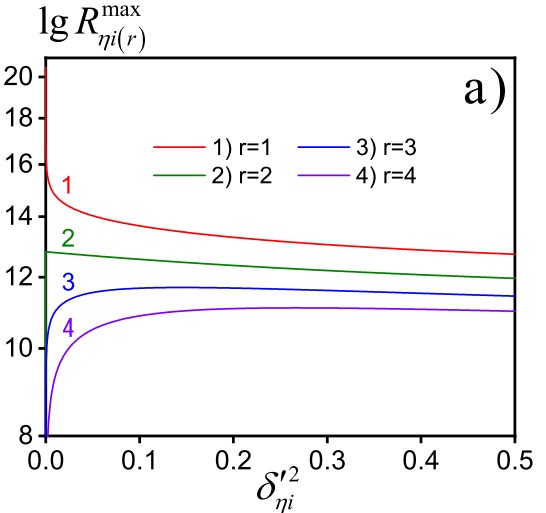
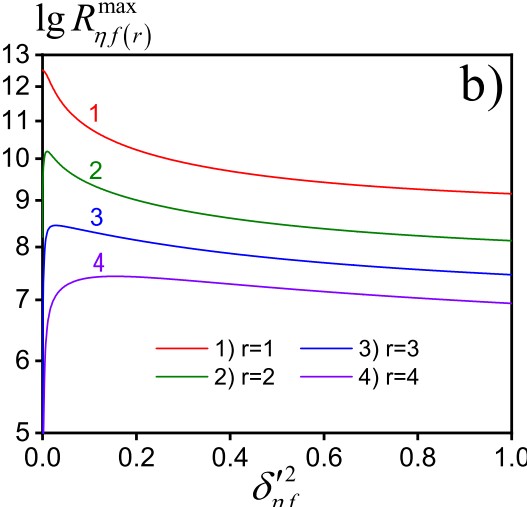

**Figure 3.** The maximum resonant differential cross-section (in units of $Z^2\alpha r_e^2$) as functions of the corresponding outgoing angles, plotted for the different values of absorbed wave photons. (**a**) $R^{\max}_{\eta i(r)}$ (21) for channel A. (**b**) $R^{\max}_{\eta f(r)}$ (22) for channel B. The frequency and intensity of the electromagnetic wave are $\omega = 20$ keV and $I = 0.746 \times 10^{27}$ Wcm$^{-2}$. The energy of the initial electrons is $E_i = 65$ GeV.

These figures differ in the frequencies ($\omega = 0.2$ keV; 20 keV) and intensities of the electromagnetic wave $\left(I = 0.746 \times 10^{23} \text{ Wcm}^{-2}; 0.746 \times 10^{27} \text{ Wcm}^{-2}\right)$. Tables 1 and 2 show the values of the resonant differential cross-sections, energies and outgoing angles of the spontaneous gamma quantum, as well as the energies of the final electron in the maximum of the corresponding distributions, corresponding to Figures 2 and 3. It can be seen from Figures 2 and 3 that at fixed frequency and intensity of the wave, the maximum value of the resonant section occurs at one absorbed photon of the wave ($r = 1$) for the zero outgoing angle $\left(\delta'^2_{\eta i} = 0, \ \delta'^2_{\eta f} = 0\right)$ and can be of the order of magnitude $10^{19} \div 10^{20} \left(Z^2\alpha r_e^2\right)$. With an increase in the number of absorbed photons of the wave, the magnitude of the maximum resonant cross-section decreases quite quickly, while still taking large values. At the same time, the position of the maxima in the corresponding distributions of resonant

cross-sections shifts towards large values of the outgoing angles of the spontaneous gamma quantum, and the energy of the spontaneous gamma quantum tends more and more to the energy of the initial electrons (see also Tables 1 and 2). Further, note that channel B is suppressed relative to channel A (34).

**Table 1.** The values of the resonant frequency and the square of the outgoing angle of the spontaneous gamma quantum for the maximum value of the maximum resonant differential cross-section (see Figure 2). The frequency and intensity of the electromagnetic wave are $\omega = 0.2$ keV and $I = 0.746 \times 10^{23}$ Wcm$^{-2}$. The energy of the initial electrons is $E_i = 65$ GeV.

| $r$ | $\delta^2_{\eta i}$ | $R^{max}_{\eta i(r)}$ $(Z^2 \alpha r_e^2)$ | $\omega'_{\eta i(r)}$ (GeV) | $E_{\eta f(r)}$ (GeV) | $\delta^2_{\eta f}$ | $R^{max}_{\eta f(r)}$ $(Z^2 \alpha r_e^2)$ | $\omega'_{\eta f(r)}$ (GeV) | $E_{\eta f(r)}$ (GeV) |
|---|---|---|---|---|---|---|---|---|
| 1 | 0 | $1.147 \times 10^{19}$ | 64.35644 | 0.64356 | 0 | $1.107 \times 10^{15}$ | 64.35 | 0.65 |
| 2 | 0.003 | $2.208 \times 10^{15}$ | 64.67565 | 0.32435 | 0.075 | $4.159 \times 10^{13}$ | 64.650625 | 0.349375 |
| 3 | 0.145 | $1.749 \times 10^{14}$ | 64.75287 | 0.24713 | 0.126 | $4.428 \times 10^{12}$ | 64.756055 | 0.243945 |
| 4 | 0.266 | $4.468 \times 10^{13}$ | 64.79493 | 0.20507 | 0.205 | $8.008 \times 10^{11}$ | 64.804155 | 0.195845 |

**Table 2.** The values of the resonant frequency and the square of the outgoing angle of the spontaneous gamma quantum for the maximum value of the maximum resonant differential cross-section (see Figure 3). The frequency and intensity of the electromagnetic wave are $\omega = 20$ keV and $I = 0.746 \times 10^{27}$ Wcm$^{-2}$. The energy of the initial electrons is $E_i = 65$ GeV.

| $r$ | $\delta^2_{\eta i}$ | $R^{max}_{\eta i(r)}$ $(Z^2 \alpha r_e^2)$ | $\omega'_{\eta i(r)}$ (GeV) | $E_{\eta f(r)}$ (GeV) | $\delta^2_{\eta f}$ | $R^{max}_{\eta f(r)}$ $(Z^2 \alpha r_e^2)$ | $\omega'_{\eta f(r)}$ (GeV) | $E_{\eta f(r)}$ (GeV) |
|---|---|---|---|---|---|---|---|---|
| 1 | 0 | $3.125 \times 10^{20}$ | 64.9934 | 0.0065 | 0 | $1.172 \times 10^{12}$ | 64.99343 | 0.00656 |
| 2 | $2 \times 10^{-5}$ | $6.343 \times 10^{12}$ | 64.9967 | 0.0032 | 0.01 | $1.534 \times 10^{10}$ | 64.99668 | 0.00331 |
| 3 | 0.145 | $4.853 \times 10^{11}$ | 64.9975 | 0.0024 | 0.028 | $2.806 \times 10^{8}$ | 64.99772 | 0.00227 |
| 4 | 0.266 | $1.239 \times 10^{11}$ | 64.9979 | 0.0020 | 0.154 | $2.681 \times 10^{7}$ | 64.99811 | 0.00188 |

It is important to emphasize that a large value of the resonant differential cross-section occurs when the resonant cross-section is integrated at the outgoing angles of the spontaneous gamma quantum relative to the momentum of the final electron (for channel A) or the initial electron (for channel B). As a result, a large order parameter $(E_i/m_*)^2 \gg 1$ appears in the resonant differential cross-section (21), (22), (25) (see also formulas (81)–(84) of the article [22]). Note that this large parameter arises due to taking into account small-order corrections in the transmitted momentum $\mathbf{q}^2$ [22].

The study provides a qualitative explanation of the possibility of obtaining high-energy gamma quanta in strong X-ray fields near neutron stars and magnetars. These high-energy gamma quanta can give production to ultrarelativistic electron–positron pairs in strong X-ray fields. As is known, there is a problem in explaining the flux of positrons (electrons) with energies of the order of 100 GeV near these objects [3,4,21]. It is a well-known phenomenon that the cosmic rays contain an anomalous abundance of high-energy positrons [3]. One might suppose that the X-ray pulsars might be the sources of these positron flows [4]. The present work allows us to give a possible explanation of these facts.

## 5. Conclusions

The study of resonant spontaneous bremsstrahlung radiation during scattering of ultrarelativistic electrons by nuclei in the field of an X-ray wave with wide frequency and intensity intervals shows:

- Under conditions when the energies of the initial electrons significantly exceed the characteristic energy of the process ($E_i \gg E_*$, see (16)), spontaneous gamma quanta will be emitted with energies close to the energy of the initial electrons (see (15) and (20)).
- The emission peaks of spontaneous gamma quanta have different outgoing angles for different numbers of absorbed photons of the wave (see Figures 2 and 3 and Tables 1 and 2).
- The resonant differential cross-section with simultaneous registration of the outgoing angles and the energy of the spontaneous gamma quantum significantly exceeds the corresponding cross-section without an external electromagnetic field. This excess can be twenty orders of magnitude for one absorbed photon of the wave.
- The resonant differential cross-section for channel A is dominant, since it exceeds the corresponding differential cross-section for channel B by more then two orders of magnitude.

The results obtained can be used to explain high-energy gamma-ray quanta near neutron stars and magnetars.

**Author Contributions:** Conceptualization, S.R.; methodology, S.R.; software, S.S.; validation, S.R., A.D. and S.S.; formal analysis, S.R. and A.D.; investigation, S.R., A.D. and S.S.; resources, A.D.; data curation, S.R. and A.D.; writing—original draft preparation, S.S.; writing—review and editing, S.R. and S.S.; visualization, S.S.; supervision, S.R. and A.D.; project administration, A.D.; funding acquisition, A.D. All authors have read and agreed to the published version of the manuscript.

**Funding:** This research received no external funding.

**Institutional Review Board Statement:** Not applicable.

**Informed Consent Statement:** Not applicable.

**Data Availability Statement:** Not applicable.

**Conflicts of Interest:** The authors declare no conflict of interest.

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
