# Peer review of "Radiation of High-Energy Gamma Quanta by Ultrarelativistic Electrons on Nuclei in Strong X-ray Fields"

_universe, doi:10.3390/universe8040218_

Round 1
Reviewer 1 Report
Authors predict the possibility of emission of high-energy photons with energies about of 100 GeV by ultrafast electrons at the bremsstrahlung during interaction with atomic nuclei in the presence of the strong X-ray fields. They suggest required experimental conditions for realization of such an experiment in the process of resonance bremsstralung. Under these conditions photons can be emitted in a narrow cone with the energies of the order of the initial ultrafast electron energy. The results can explain high energy photons near pulsars.
Author Response
We are grateful to the reviewer for the positive review.
Reviewer 2 Report
The authors of the work "Radiation of high-energy gamma quanta by ultra-relativistic electrons on nuclei in strong X-ray fields" demonstrate the possibility of O(100) GeV electron-to-photon conversion with the aid of resonant spontaneous bremsstrahlung. They show that in a special set of the laser facility with unprecedented intensity, a few number of photon absorption can induce a large rate of photon emission collinear to the direction of the electron, taking a large fraction of the electron's energy.
This work may be considered for acceptance if the authors can address the following questions, improve on the delivering, and provide more details on specific formulas and definitions.
Here are the questions organized by sections:
For Section I: Introduction
1) Is the resonance phenomena highly relies on the monochromatic and coherent electron-magnetic field? As I thought this condition to be hard to exist in astrophysical environment.
2) Please define channel A and B or referring to figure one around line 37-38.
For Section 2: Resonant Frequency in a Strong Field
1) Please explain the use of quasimomenta (Is equation 8 the definition of the quasimomentum?) and define the four-momentum production notation, e.g., (pk).
2) If the author is focusing on the resonant region where the intermediate electron goes on-shell, then the electron can in principle interact with multiple scattering centers if the medium is dense. Are there any restrictions to the medium density such that it is sufficient to only consider a single scattering with one nucleus?
For section 3: The Maximum Resonant Differential SB Cross-Sections
1) Though the magnitude of the resonant cross-section can be huge, as shown in section four, it is only achieved at a special angle and special x' = omega/E. I wonder if the effect is still large when consider angular-integrated and x'-integrated spectrum.
For section 4: Results
1) A question about the zero-degree emission. Usually, in the vacuum, the zero-degree emission is divergent due to the collinear divergence of photon emission and is usually screened by either medium effects or simply cut by detector resolution. In the current setting of this work, I guess it is some resonance/damping effect that regulate the collinear radiation? Can the authors clarify this point?
Furthermore, However, the author mentioned that the resonant cross-section can be 10^20 larger than the non-resonant one. However, in figures 2 and 3, the spectrum is plotted in units of Z^2 alpha r_e^2. The latter does not contain the essential angular and longitudinal structures of non-resonant radiation spectrum. Is this the proper quantity to be used to scale the resonant calculation?
2) Please center table 2.
For section 5: Conlusion
1) The last sentence, "The results obtained can be used to explain high-energy gamma-ray quanta near neutron stars and magnetars." Is it possible for the author to plug in an estimated number of the magnetic fields, and compute the spectrum of the resonant radiation averaging over the angles and other parameters, such as the angle between the electron and the magnetic field.
Author Response
"Please see the attachment."

Reviewer 3 Report
Report on “Radiation of High-energy Gamma Quanta…” by S. Roshchupkin et al.
The authors study theoretically bremsstrahlung in presence of an external laser field, and derive the resonance frequencies and the cross section at resonance.
I must admit that I do not really see the significance of the results presented in the manuscript. First, the process considered is resonant so that the bremsstrahlung process in the external field separates into (1) emission of a photon by the nonlinear Compton effect and (2) scattering on a nucleus without emission of radiation. It seems that all the relevant cross sections and photon energies are the same as for nonlinear Compton scattering, which has been thoroughly studied in numerous previous publications, and that the present manuscript does not provide any new information. What is the significance of including the interaction with the nucleus here?
The statement that the “cross section exceeds the corresponding cross section without an external field by twenty orders of magnitude” seems to mean simply that the probability of nonlinear Compton scattering in a very intense field is much larger that the probability of bremsstrahlung.
Second, it is written that “the results obtained can be used to explain high-energy gamma-ray quanta near neutron start and magnetars”. This connection to neutron stars and magnetars is not at all clear, and no discussion is presented in the manuscript. Do strong, periodic X-ray fields exist around neutron stars? Can we find 100 GeV electrons near neutron stars? What are the density distributions and the energy distributions of such electrons? By estimating the spectrum of the emitted radiation, does it agree with the experimentally observed spectra? This kind of considerations are completely absent in the manuscript.
I cannot recommend publication of the manuscript unless the above comments have been addressed. I believe that this would require a substantial revision of the manuscript.
Author Response
"Please see the attachment."

Round 2
Reviewer 2 Report
The authors have responded to my comments and suggestions in a satisfactory manner, and the manuscript can be accepted for publication.
In the future, I would be really interested in following your developments if you can put this results into real estimation of the production spectra of the cosmic high energy particles.
Author Response
We are grateful to the reviewer for the positive feedback.
Reviewer 3 Report
I do not believe that the authors have answered my questions appropriately.
There is no adequate explanation why a sequential process of first Compton scattering and then scattering at a proton would enhance the cross section by 20 orders of magnitude. No explanation of this fact has been included in the manuscript.
On the topic of the relation to astrophysical phenomena, the authors have included a statement "there is a problem in explaining the flux of positrons (electrons) with energies of the order of 100 GeV near these objects", but the theory in the manuscript does not deal with positron production or positron acceleration. I wanted to know if periodic xray fields of the intensities considered in the manuscript exist in space. In X-ray pulsars mentioned in the manuscript, what is the typical wavelength? Intensity? Can these Xray sources be considered as coherent, so that the theory in the manuscript is valid? Also, I wanted to see an estimate of the spectrum of gamma rays given a realistic X-ray source found in space. No considerations like this have been added to the manuscript.
I do not recommend publication of the manuscript.
Author Response
"Please see the attachment."
